# Self-Polish: Enhance Reasoning in Large Language Models via Problem Refinement

**Zhiheng Xi**[1*], **Senjie Jin**[1*], **Yuhao Zhou**[1], **Rui Zheng**[1], **Songyang Gao**[1],
**Tao Gui**[2†] , **Qi Zhang**[1†], **Xuanjing Huang**[1]

[1] School of Computer Science, Fudan University, Shanghai, China

[2] Institute of Modern Languages and Linguistics, Fudan University, Shanghai, China

{zhxi22,sjjin22,zhouyh21,gaosy21}@m.fudan.edu.cn ,

{rzheng20,tgui,qz,xjhuang}@fudan.edu.cn

## Abstract

To enhance the multi-step reasoning capabilities of large language models, researchers have extensively explored prompting methods, notably the Chain-of-Thought (CoT) method which explicitly elicits human-like rationales. However, they have inadvertently overlooked the potential of enhancing model reasoning performance by formulating higher-quality problems [1]. In this work, we start from the problem side and propose Self-Polish (SP), a novel method that facilitates the model's reasoning by guiding it to progressively refine the given problems to be more comprehensible and solvable. We also explore several automatic prompting varients and propose the Self-Polish prompt bank for the community. SP is orthogonal to all other prompting methods of answer/reasoning side like CoT, allowing for seamless integration with state-of-the-art techniques for further improvement. Thorough experiments show that the proposed method attains notable and consistent effectiveness on five reasoning benchmarks across different models. Furthermore, our method also showcases impressive performance on robustness evaluation. Codes and prompts are available at https://github.com/WooooDyy/Self-Polish.

## 1 Introduction

Large language models (LLMs) have achieved impressive performance on a variety of NLP tasks (Brown et al., 2020; Otter et al., 2021; Chowdhery et al., 2022), but their capability to perform multi-step reasoning is considered a limitation, which can not be tackled solely by scaling up the model size (Rae et al., 2021; Srivastava et al., 2022). To address this challenge, many prompting methods have been proposed to elicit reasoning in LLMs, and

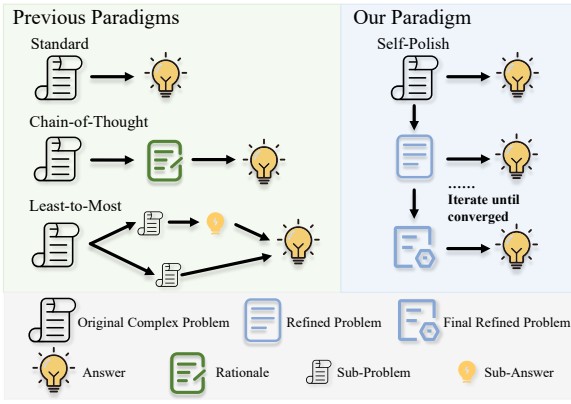

Figure 1: Schematic comparison between Self-Polish and other representative approaches for reasoning with prompting. Previous paradigms enhance the reasoning capability of LLMs from the aspect of the **answer side/reasoning side**, while our method starts from the **problem side**, and refines problems to be simpler and more comprehensible for models.

have demonstrated significant effectiveness (Wei et al., 2022b; Fu et al., 2022; Zhou et al., 2022a).

Chain-of-Thought (CoT) is a breakthrough method that teaches a language model to imitate the step-by-step reasoning process of humans to solve a reasoning task (Wei et al., 2022b). Many following work has explored variants of CoT to improve the quality of rationales of LLMs (Kojima et al., 2022; Fu et al., 2022; Zhou et al., 2022a). There is also a line of work that optimizes the rationales for better consistency and continuity (Wang et al., 2022; Li et al., 2022; Zelikman et al., 2022; Zheng et al., 2023), and a representative one is Self-Consistency (SC). SC generates diverse reasoning paths and answers, and then leverages the majority vote strategy to get the most consistent answer (Wang et al., 2022). Despite the boosted reasoning performance of the aforementioned methods, they focus on the **answer/reasoning side**, and little emphasis has been placed on the **problems side**.

Actually, the clarity and logical structure of the

---

*Equal contribution.

†Corresponding author.

[1]A reasoning problem often consists of two parts: the context and the final question (Creswell et al., 2022).

**problem** description are crucial factors for human understanding and model comprehension (Shou and Smithson, 2015; Faruqui and Das, 2018; Chu et al., 2020). LLMs often exhibit poor reasoning performance when confronted with low-quality real-world reasoning problems, which may be excessively long, ambiguous, unclear in focus, or contain irrelevant information (Zellers et al., 2018; Shi et al., 2023; Ye and Durrett, 2022). To tackle this challenge, we consider refining problems into a better formulation.

In this work, we propose Self-Polish (Figure 1 right) that leverages LLMs themselves to refine reasoning problems without training for better reasoning performance. We first present several principles for refined problems: concise, clear, well-focused, and absent of irrelevant information. To achieve our goal, we propose the Self-Polish Prompt Bank which includes several feasible solutions as outlined in the following text. An intuitive strategy is to reformulate problems via instruction-following (Sanh et al., 2022; Ouyang et al., 2022), and we call it **zero-shot problem refining**. Next, we include demonstrations in the prompts (Brown et al., 2020; Chowdhery et al., 2022) to enable models to better internalize and apply the principles, which is defined as **in-context problem refining**. During the construction of the demonstrations, we incorporated a curated collection of problem-refining patterns, e.g., eliminating irrelevant information, rearranging the logic structure, and organizing local conditions into new ones in parallel. Moreover, we explore automatic prompting methods to construct enhanced prompts and mitigate manual efforts, based on the criteria of complexity (**complexity-based Self-Polish**) or diversity (**automatic Self-Polish**). To further enhance the reliability and consistency of the generated problems, we propose to progressively refine problems until obtaining a convergent answer.

Experiments show that our method consistently improves the reasoning performance of various models (i.e., Text-davinci-002, Text-davinci-003, GPT-3.5-Turbo) on five benchmarks (Table 1 & Figure 3). Moreover, the proposed method is orthogonal to all other reasoning-side state-of-the-art prompting methods, making it convenient to be combined with them for further improvement. Detailed experiments demonstrate that the performance of reasoning-side methods can be significantly boosted when integrated with SP (Table 2 &

Table 3). Self-Polish also showcases exceptional performance on robustness evaluation (Figure 4).

In summary, we make the following contributions:

1. We propose a novel method, Self-Polish, to improve the reasoning performance and robustness of LLMs.

2. We demonstrate the effectiveness of our method when applied alone or combined with other prompting approaches on five benchmarks with different models.

3. We believe that the proposed Self-Polish represents an important step in enhancing LLMs' reasoning capabilities by shifting the perspective from the answer/reasoning side to the problem side. We hope it could inspire future research in this field.

## 2 Related Work

**Multi-step reasoning.** Multi-step reasoning tasks have posed significant challenges for language models (Rae et al., 2021; Bommasani et al., 2021; Qiao et al., 2022), and it is considered as an emergent ability of LLMs (Schaeffer et al., 2023). It is in these tasks that the effectiveness of few-shot prompting begins to surpass that of full training set fine-tuning (Lewkowycz et al., 2022). Moreover, such capability is considered important in building more complex artificial intelligence such as large language model-based agents (LLM-based agents) (Xi et al., 2023). Our work represents a significant stride in enhancing the ability of language models to perform multi-step reasoning tasks, through the facilitation of models' comprehension and processing of given reasoning problems.

**Reasoning with prompting.** Prompting strategies have substantially improved the reasoning ability of LLMs by a large margin (Qiao et al., 2022; Lewkowycz et al., 2022). An important line of work in this area is Chain-of-Thought (CoT) prompting which elicits the reasoning ability of models by prompting them to imitate the step-by-step reasoning process of humans (Wei et al., 2022b; Kojima et al., 2022; Fu et al., 2022; Zhou et al., 2022a). Another line of work focuses on optimizing the rationales for better consistency and continuity (Wang et al., 2022; Li et al., 2022; Zelikman et al., 2022; Zheng et al., 2023). A representative one is Self-Consistency (SC), which samples

**[Original Problem₀]** : Kylie makes 10 beaded necklaces on Monday and 2 beaded necklaces on Tuesday. Then Kylie makes 5 beaded bracelets on Wednesday. 20 beads are needed to make one beaded necklace. 10 beads are needed to make one beaded bracelet. Ada bought 2000 tomatoes from the grocery store. How many beads does Kylie use in total to make her jewelry?
**[Answer₀]** : 120. 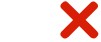

**Begin**, Problem Refine (Remove Irrelevant Information)

**[Refined Problem₁]** : Kylie makes 12 beaded necklaces, 5 beaded bracelets. Each beaded necklace needs 20 beads. Each beaded bracelet requires 10 beads. How many beads does Kylie use in total to make her jewelry?
**[Answer₁]** : 155. 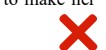

A₁ != A₀, Continue Refine (Reorder Conditions)

**[Refined Problem₂]** : Kylie makes 12 beaded necklaces, and each beaded necklaces needs 20 beads. She also makes 5 beaded bracelets, and each beaded bracelet needs 10 beads. How many beads does Kylie use in total to make her jewelry?
**[Answer₂]** : 290. ✓

A₂ != A₁, Continue Refine (Summary Local Conditions)

**[Refined Problem₃]** : Kylie requires 240 beads to make beaded necklaces. She also requires 50 beads to make beaded bracelets. How many beads does Kylie use in total to make her jewelry?
**[Answer₃]** : 290. ✓

A₃ == A₂, Return A₃

Figure 2: An example illustrating the framework and problem-refining patterns of Self-Polish. In the first re-fining iteration, the irrelevant information "Ada bought 2000 tomatoes from the grocery store." is removed. In the second iteration, the conditions are reordered for easier calculation of the number of beads required for each type of beaded product. In the third iteration, the local conditions were parallelly combined to form new conditions (the total number of beads required for neck-laces and bracelets).

multiple reasoning paths and generate the most consistent answer by majority vote (Wang et al., 2022). Different from Self-Polish, the aforementioned strategies emphasize improving the quality of rationales from the answer/reasoning side. Our method is a problem-side method, so it is orthogonal to all of them and can be combined with them for further improvement.

See Appendix A for more related work and the detailed differences between Self-Polish and Least-to-Most (Zhou et al., 2022a).

## 3 Self-Polish Prompting

In this section, we first revisit previous prompting paradigms aiming at solving reasoning problems. Next, we describe the proposed Self-Polish method detailedly.

### 3.1 Revisiting Paradigms of Reasoning Problem Solving

In the context of enhancing the capabilities of LLMs, the prompting technique has emerged as one of the most popular approaches owing to its training-free nature and effectiveness (Qiao et al., 2022; Lewkowycz et al., 2022). Here, we formal-ize several representative paradigms. See Figure 1 for a schematic comparison between them and our method.

**Standard.** The prompt contains $k\times$ [Problem, Answer] pairs, followed by the test problem.

**Chain-of-Thought (Wei et al., 2022b).** The prompt contains $k\times$ [Problem, Rationale, Answer] tuples, followed by the test problem. This method teaches models to generate rationales and answers, achieving significant improvement in reasoning. **Auto-CoT** (Fu et al., 2022) and **Complex-CoT** (Zhou et al., 2022a) are two automatic varients that constructs CoT demonstrations according to the criteria of problem diversity and reasoning com-plexity, respecticely.

**Least-to-Most (Zhou et al., 2022a).** The mod-els are taught to first reduce problems into sub-problems and then solve them sequentially. There are two kinds of prompts. The first is the prob-lem reduction prompt that contains $m\times$ [Original Problem, Sub-Problems] pairs, followed by the test original problem. The second is the problem-solving prompt that contains $k\times$ [Original Problem, $n\times$ (Sub-Problem, Sub-Answer)] tuples, followed by the test original problem and the current sub-problem to solve.

**Summary.** All previous methods focus on the answer/reasoning side, and it is convenient to com-bine them with Self-Polish which puts emphasis on the problem side.

### 3.2 Problem-Refining Prompting

#### 3.2.1 Refining Principles

We expect the newly generated problems to be eas-ier to understand and process, so they should adhere to the following principles: (1) **conciseness**, the problems should not be overly long, ensuring they remain easily understandable; (2) **clarity**, the prob-lems should avoid ambiguous phrasing and instead utilize quantitative representations (e.g., Arabic nu-merals) whenever possible; (3) **focus**: the problems should clearly convey the intended subject matter,

| METHOD | Progressively | DATASET | | | | | AVERAGE |
|---|---|---|---|---|---|---|---|
| | | GSM8K | AQuA | SVAMP | MultiArith | MathQA | |
| Standard | ✗ | 15.8 | 28.3 | 72.9 | 35.1 | 28.2 | 36.1 |
| +Zero-shot SP | ✗ | 22.4(↑ 6.6) | 28.3(0) | 73.2(↑ 0.3) | 43.1(↑ 8.0) | 25.4(↓ 2.8) | 38.5(↑ 2.4) |
| | ✓ | 24.0(↑ 8.2) | 28.7(↑ 0.4) | 72.2(↓ 0.7) | 51.7(↑ 16.6) | 26.8(↓ 1.4) | 40.7(↑ 4.6) |
| +In-context SP | ✗ | 24.3(↑ 8.5) | 30.3(↑ 2.0) | 73.9(↑ 1.0) | 50.6(↑ 15.5) | 29.4(↑ 0.8) | 41.7(↑ 5.6) |
| | ✓ | 25.3(↑ 9.5) | 29.5(↑ 1.2) | 73.9(↑ 1.0) | 52.9(↑ 17.8) | 28.6(↑ 0.4) | 42.0(↑ 5.9) |
| +Auto-SP | ✗ | 24.3(↑ 8.5) | 29.9(↑ 1.6) | 72.6(↓ 0.3) | 54.0(↑ 18.9) | 27.6(↓ 0.6) | 41.7(↑ 5.6) |
| | ✓ | 24.3(↑ 8.5) | 30.3(↑ 2.0) | 72.9(0) | 56.7(↑ 21.6) | 28.2(0) | 42.5(↑ 6.4) |
| +Complex-SP | ✗ | 23.0(↑ 7.2) | 29.9(↑ 1.6) | 73.2(↑ 0.3) | 52.3(↑ 17.2) | 29.6(↑ 1.4) | 41.6(↑ 5.5) |
| | ✓ | 24.6(↑ 8.8) | 28.7(↑ 0.4) | 72.9(0) | 55.7(↑ 20.6) | 30.0(↑ 1.8) | 42.4(↑ 6.3) |

Table 1: Evaluating different strategies of the Self-Polish prompting bank on several benchmarks. Performance gains/drops are highlighted with green/red. The results are with Text-davinci-003. "Progressively" represents whether using the progressively refining framework in Section 3.3.

making it evident what the question is asking; (4) **absence of irrelevant information**: the problems should be free from extraneous details that could cause confusion or distractions.

### 3.2.2 Construction of Refining Prompts

**Zero-shot Self-Polish.** It is difficult to internalize the aforementioned principles within the model via training due to the tedious process of constructing a corresponding dataset and potential catastrophic forgetting problems (Goodfellow et al., 2014; Parisi et al., 2019). So we turn to training-free strategies.

As LLMs demonstrate emergent abilities of instruction-following (Schaeffer et al., 2023; Sanh et al., 2022; Wei et al., 2022a), a simple and intuitive strategy to refine problems is prompting LLMs with an instruction. In the instruction, we guide the model to rewrite new versions of the original reasoning problem to be more understandable and easy to answer, and never omit any useful information. The prompt contains [Instruction, Original Problem] and the model responds with a newly generated problem. Next, we can adopt any prompting method in Section 3.1 to get the answer to the new problem, and we take this answer as the final one. We conduct preliminary validation experiments and the results are illustrated in Table 1. Zero-shot refining can consistently improve reasoning performance on various benchmarks.

**In-context Self-Polish.** As empirical results show that zero-shot refining can only provide limited performance gain, especially on difficult datasets, we then add demonstrations to the prompt to enable models to better internalize and apply

design principles. Specifically, demonstrations are formulated as [Original Problem, New Problem] pairs, and we incorporate a curated collection of **problem-refining patterns** in the demonstrations: (1) remove irrelevant information, as the first iteration in Figure 2; (2) rearrange the logic structure and group relevant conditions together to better match the reasoning logic of the model, as the second iteration in Figure 2; (3) summarize local conditions into new ones in parallel, as the third iteration in Figure 2.[2] Results in Table 1 show that in-context problem refining yields more performance gain than zero-shot refining.

**Automatic Self-Polish.** This is an automatic variant of the in-context problem-refining. We draw inspiration from Zhang et al. (2022) and construct the refining prompt according to the diverse semantics of problems with the technique of $k$-means clustering. The underlying hypothesis is that a diverse set of demonstrations can cover a broad semantic space of problems, thereby the model can locate relevant reference demonstrations for more test examples. Table 1 shows that Auto-SP also yields significant improvement.

**Complexity-based Self-Polish.** This is another variant of the in-context problem-refining for automatically selecting refining demonstrations. We draw inspiration from Fu et al. (2022) and construct the refining prompt according to the complexity of each problem. The underlying hypothesis is that

---

[2]Note that a single example typically does not encompass all refining strategies. The example is constructed solely to illustrate our design patterns.

the refining ability of the model can generalize from complex problems to simpler ones. Table 1 demonstrates that Complex-SP can also yield substantial performance gain.

### 3.3 Progressively Refining Framework

To enhance the consistency and reliability of the refined problems, we propose a progressive framework that has two stages: the problem-solving stage (Section 3.1) and the problem-refining stage (Section 3.2). The two stages are executed alternatively until the return condition is satisfied.

**Return condition & answer selection.** There are two situations that terminate the iterative process. The first is when the last two answers are the same, indicating convergence of the answer. In this case, we can directly return the answer. The second situation is when the iteration number exceeds the maximum count $T = 2$.[3] In such case, we have multiple options for selecting the final answer, such as the answer to the original problem, the answer to the first generated problem, the answer to the last generated problem, or utilizing a majority voting approach to select the answer (Wang et al., 2022), which will be discussed in our ablation study in Section 5.1. Here we choose the answer to the last generated problem by default. As shown in Table 1, adding Progressively Refining to our method can bring further improvement across different prompt-construction approaches.

The overall framework is shown in Algorithm 1 in Appendix B.

## 4 Experiments

In this section, we conduct experiments to demonstrate the effectiveness and robustness of SP.

### 4.1 Experimental Setups.

**Models.** We employ three GPT-series models, namely text-davinci-002, text-davinci-003, and GPT-3.5-Turbo (Brown et al., 2020; Ouyang et al., 2022), as they are widely recognized and accessible to the public, ensuring reproducibility of our research. Our experiments are based on OpenAI's API. All methods use greedy decoding (i.e., $temperature = 0$) for stable responses.

---

[3] One iteration means one time of problem refinement. Note that a bigger $T$ can yield a larger performance gain, as discussed in Section 5.1 Here we set $T = 2$ to achieve a balance in computational efficiency and performance.

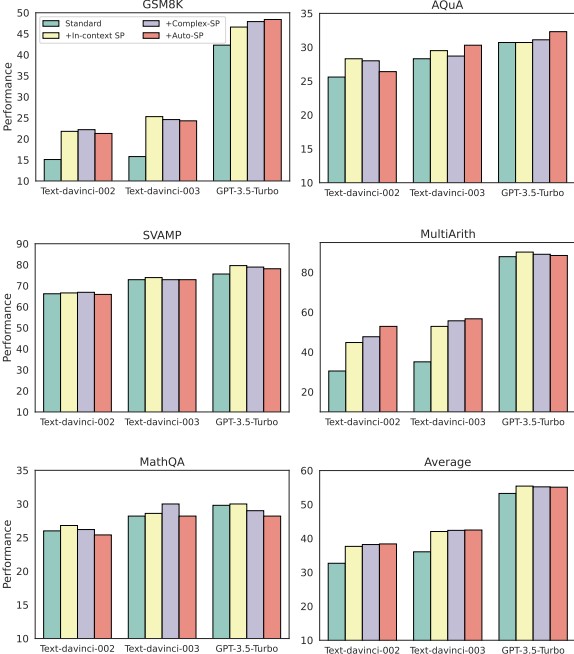

Figure 3: Evaluating Self-Polish on various benchmarks with different models. Self-Polish consistently improves reasoning performance across multiple models and benchmarks.

**Datasets.** We evaluate the performance of our method on five reasoning datasets, including GSM8K (Cobbe et al., 2021), AQuA (Ling et al., 2017), SVAMP (Patel et al., 2021), MultiArith (Roy and Roth, 2015) and MathQA (Amini et al., 2019). The datasets are evaluated by prior studies in the field of multi-hop reasoning (Wei et al., 2022b; Fu et al., 2022; Zhou et al., 2022a). We evaluate on the whole test set of AQuA and GSM8K. For other datasets, we adopt the split from Mishra et al. (2022) or randomly select 500 test instances, and perform 3 restarts for stable results.

**Prompts.** For the sake of generalizability, GSM8K, SVAMP and MultiArith share the same Self-Polish prompts constructed from GSM8K; AQuA and MathQA share the same Self-Polish prompts constructed from AQuA. See Appendix F for SP prompts. The prompts for the standard few-shot prompting method are from Wei et al. (2022b). The prompts for Chain-of-thought, Least-to-Most, Auto-CoT and Complex-CoT are from previous work (Wei et al., 2022b; Zhou et al., 2022a; Zheng et al., 2023; Fu et al., 2022; Zhang et al., 2022). While prompts of LtM are not available for some datasets, we manually construct them.

See more implementation details in Appendix C.

| Answer Side | Problem Side | Dataset | | | | | Average |
|---|---|---|---|---|---|---|---|
| | | GSM8K | AQuA | SVAMP | MultiArith | MathQA | |
| Chain-of-Thought | No Refinement | 56.1 | 44.9 | 80.3 | 90.8 | 41.0 | 62.6 |
| | In-context SP | 56.7(↑ 0.6) | 48.8(↑ 3.9) | 81.6(↑ 1.3) | 88.5(↓ 2.3) | 43.0(↑ 2.0) | 63.7(↑ 1.1) |
| | Auto-SP | 56.9(↑ 0.8) | 49.2(↑ 4.3) | 78.3(↓ 2.0) | 89.7(↓ 1.1) | 42.8(↑ 1.8) | 63.4(↑ 0.8) |
| | Complex-SP | 58.1(↑ 2.0) | 47.3(↑ 2.4) | 81.3(↑ 1.0) | 90.2(↓ 0.6) | 43.2(↑ 2.2) | 64.0(↑ 1.4) |
| Least-to-Most | No Refinement | 59.3 | 42.1 | 82.9 | 83.9 | 39.0 | 61.4 |
| | In-context SP | 61.2(↑ 1.9) | 44.1(↑ 2.0) | 84.9(↑ 2.0) | 85.1(↑ 1.2) | 41.2(↑ 2.2) | 63.3(↑ 1.9) |
| | Auto-SP | 61.6(↑ 2.3) | 44.9(↑ 2.8) | 82.9(0) | 83.9(0) | 41.2(↑ 2.2) | 62.9(↑ 1.5) |
| | Complex-SP | 62.9(↑ 3.6) | 47.6(↑ 5.5) | 84.2(↑ 1.3) | 86.2(↑ 2.3) | 40.6(↑ 1.6) | 64.3(↑ 2.9) |
| Auto-CoT | No Refinement | 59.4 | 46.5 | 75.6 | 92.5 | 41.4 | 63.1 |
| | In-context SP | 59.4(0) | 47.2(↑ 0.7) | 77.6(↑ 2.0) | 90.6(↓ 1.9) | 45.8(↑ 4.4) | 64.1(↑ 1.0) |
| | Auto-SP | 60.5(↑ 1.1) | 48.0(↑ 1.5) | 75.6(0) | 89.7(↓ 2.8) | 44.4(↑ 3.0) | 63.6(↑ 0.5) |
| | Complex-SP | 60.1(↑ 0.7) | 50.8(↑ 4.3) | 78.2(↑ 2.6) | 93.1(↑ 0.6) | 43.0(↑ 1.6) | 65.0(↑ 1.9) |
| Complex-CoT | No Refinement | 67.5 | 47.6 | 78.3 | 94.3 | 43.6 | 66.3 |
| | In-context SP | 66.2 (↓ 1.3) | 50.8(↑ 3.1) | 80.9(↑ 2.6) | 90.8(↓ 3.5) | 44.6(↑ 1.3) | 66.7(↑ 0.4) |
| | Auto-SP | 68.7(↑ 1.2) | 45.7(↓ 1.9) | 78.3(0) | 92.0(↓ 2.3) | 42.4(↓ 1.2) | 65.4(↓ 0.9) |
| | Complex-SP | 68.5(↑ 1.0) | 50.4(↑ 2.8) | 78.6(↑ 0.3) | 93.1(↓ 1.2) | 44.8(↑ 1.2) | 67.1(↑ 0.8) |

Table 2: Evaluation results when combining Self-Polish with other answer/reasoning side prompting strategies. The results are with Text-davinci-003. The best performance for each answer side strategy of one task is underlined. The best performance for each task is in **bold**.

## 4.2 Experimental Results

**Standard few-shot setting.** Figure 3 shows the results of evaluating the performance in the standard few-shot setting. We can find that : (1) Our method consistently improves reasoning performance by a large margin across multiple models and datasets, indicating its capability to enhance model understanding of problems. (2) On relatively weaker models, automated prompting methods like Auto-CoT and Complex-CoT yield more gains compared to in-context SP. However, on stronger models, the differences in performance gain between the three approaches are not significant, revealing that the stronger models are less sensitive to prompts.

**Combining Self-Polish with other prompting strategies.** Table 2 demonstrates the evaluating results when combining our method with other state-of-the-art reasoning-side promoting strategies. There are several critical and interesting observations: (1) Generally, SP yields substantial performance gains for all reasoning-side methods, revealing that when the model is able to better comprehend problems, both its step-by-step reasoning capabilities and problem decomposition abilities can be significantly enhanced. (2) Whether for the reasoning side or the problem side, the Complex-based approach performs the best. This indicates that LLMs have the ability to generalize from com-

plex tasks to simple ones, both in terms of reasoning and problem refinement. (3) As Fu et al. (2022) stated, the average number of words in problems, i.e., GSM8K (46.9), AQuA (51.9), SVAMP (32.1), MultiArith (31.2), and MathQA (60.1), can serve as a proxy for measuring the reasoning complexity of each task. We find that the more challenging the task, the higher the improvement achieved by SP, highlighting its suitability for intricate reasoning tasks. It is noteworthy that when combined with the CoT-series methods, our approach has limited improvement on MultiArith. This could be because the task itself can already be well solved by CoT and is relatively simple. Excessive refinement of simple problems carries the risk of information loss or semantic alterations, leading to a decline in performance, as depicted in Figure 9.

**Robustness evaluation.** GSM-IC (Shi et al., 2023) is an adversarial arithmetic reasoning dataset with distracting information in the problem to fool the model. So it is well-suited for evaluating the robustness of models. It has two splits: GSM-IC-2step which contains problems that require two reasoning steps to solve and GSM-IC-mstep which contains problems that require more than two reasoning steps to solve. As shown in Figure 4, our method enhances the robustness and reliability of various models across different prompting techniques, shielding them from the interference of low-quality problems.

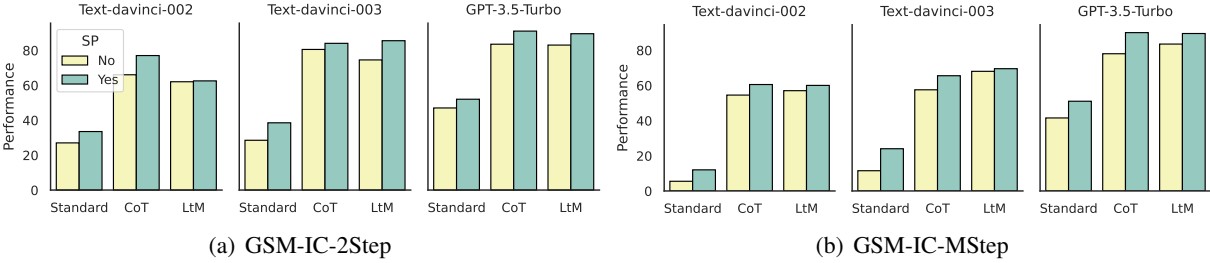

(a) GSM-IC-2Step

(b) GSM-IC-MStep

Figure 4: Evaluation results on GSMIC (Shi et al., 2023). Self-Polish (SP) enhances the robustness and reliability of various models when combined with different prompting techniques.

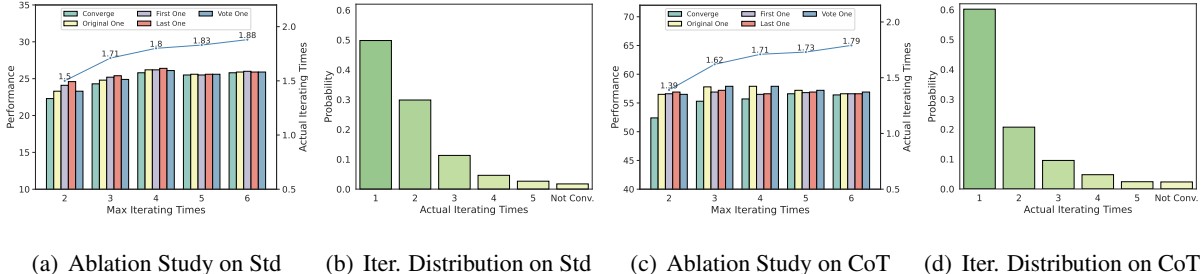

(a) Ablation Study on Std  (b) Iter. Distribution on Std  (c) Ablation Study on CoT  (d) Iter. Distribution on CoT

Figure 5: Ablation studies and the distribution of actual iterating times. (a) and (c) illustrate the performance (vertical axis on the left) when using different final answer selection strategies and different max iterating times $T$. The "Converge" means the performance calculated by $N_{conv}/N_{all}$ where $N_{conv}$ means the number of examples that are answered correctly with converged answers, while the $N_{all}$ means the number of all test examples. We also incorporate a line to represent the average actual iteration times at each value of $T$ (vertical axis on the right). In (b) and (d), we show the distribution of actual iterating times when we set $T = 5$.

## 5 Discussion

### 5.1 Ablation Studies

As mentioned in Section 3.3, the maximum iteration times $T$ and the strategy to select the final answer if the convergence is not achieved are two main components of Self-Polish. Here we perform ablation studies on them.

**Max iterating times $T$.** As shown in Figure 5(a) and Figure 5(c), for both the Standard and CoT methods, larger iteration counts lead to higher convergence accuracy ("Converge" in figures), which aligns with common knowledge and further demonstrates the effectiveness of our method: by gradually optimizing problems, we enable the model to handle them more easily. But when $T$ is too big, the performance of SP may suffer a drop, indicating that excessive rewriting can lead to a decline in the quality of problems. We set $T = 2$ not only for the sake of efficiency, but also because it can achieve competitive performance especially when combined with CoT-series methods.

**Final answer selection strategies.** We can easily observe that with a smaller $T$, the "Last One" strategy tends to have an advantage, while as the iteration count increases, other strategies become more effective, even outperforming "Last One". This is intuitive as after multiple rewriting iterations, the semantic meaning of a problem may deviate significantly from the original one.

### 5.2 Analysis of Actual Iterating Times

Figure 5(a) and Figure 5(c) show that the actual iterating times $T_{actual}$ does not grow significantly as the max iterating times $T$ increases, revealing that SP can achieve a converged answer on most of the problems with few iterations. To verify this, we illustrate the distribution map of $T_{actual}$ with $T = 5$ in Figure 5(b) and Figure 5(d). $T_{actual}$ exhibits a long-tail distribution, with only a few samples exceeding the max times. This finding provides evidence that our method is highly efficient that consumes few additional computational resources.

### 5.3 Further Improvement for Self-Consistency

Self-Consistency is a prompting method that samples multiple reasoning paths and generates a consistent answer by majority vote strategy (Wang

| METHOD | SC Path | Self-Polish | DATASET | |
|--------|---------|-------------|---------|---------|
| | | | AQuA | MathQA |
| Auto-CoT | 5 | ✗ | 61.8 | 65.2 |
| | 5 | ✓ | 66.1 | 68.4 |
| | 10 | ✗ | 61.4 | 67.4 |
| | 10 | ✓ | 65.0 | 71.0 |
| | 20 | ✗ | 63.0 | 69.0 |
| | 20 | ✓ | **68.1** | **72.0** |
| Complex-CoT | 5 | ✗ | 61.4 | 61.6 |
| | 5 | ✓ | 65.0 | 64.4 |
| | 10 | ✗ | 65.4 | 62.4 |
| | 10 | ✓ | 66.1 | 64.8 |
| | 20 | ✗ | 65.4 | 64.0 |
| | 20 | ✓ | **67.0** | **65.6** |

Table 3: Evaluation results of combining Self-Polish with Self-Consistency on GPT-3.5-Turbo. In the problem side, we use the Complex-SP. The best results in each manner are highlighted in **bold**. With Self-Polish, Self-Consistency performs better.

et al., 2022). It has proved effective in various reasoning benchmarks (Wang et al., 2022). Here, we combine the Self-Polish and Self-Consistency methods to investigate whether there will be further performance improvement. We conduct experiments on two difficult datasets (i.e., GSM8K and AQuA) with $temperature = 0.7$ for diversity following (Wang et al., 2022).

Results in Table 3 demonstrate that SP provides a substantial performance gain for SC in Auto-CoT and Complex-CoT manners. Moreover, an increase in the number of reasoning paths leads to a corresponding improvement in performance, showing the advantage of voting strategy.

## 5.4 Case Study

To further demonstrate the effectiveness of the problem-refining patterns we proposed and how our method embodies the proposed principles, we conducted a case study as shown in Figure 6. More cases can be found in the Appendix D (Figure 7 and Figure 8).

From Figure 6, we observe that removing irrelevant information (i.e., "Grover's neighbor made a salary of $10 last year.") can help the model avoid distractions and facilitate accurate reasoning. Next, rearranging the problem conditions and grouping pertinent conditions together can facilitate the model in generating more effective novel deductions during the process of reasoning (e.g., resulting in the streamlined computation of the total number of face masks in Refined Problem$_2$).

**[Original Problem$_0$]** : Grover bought 3 boxes of face masks. He plans to sell them for $0.50 each. Grover's neighbor made a salary of $10 last year. If each box has 20 face masks, and Grover bought the 3 boxes for $15, how much will be his total profit?
**[Rationale$_0$]** : Grover bought 3 boxes of face masks. He plans to sell them for $0.50 each. So he will make 3 * 0.50 = 1.50 dollars in total. Grover's neighbor made 10 dollars last year. So Grover's profit will be 1.50 - 10 = -8.50. The answer is -8.50.
**[Answer$_0$]** : -8.50. ❌

--------
**Begin**, Problem Refine (Remove Irrelevant Information)
**[Refined Problem$_1$]** : Grover bought 3 boxes of face masks. He plans to sell them for $0.50 each. If each box has 20 face masks, and Grover bought the 3 boxes for $15, how much will be his total profit?
**[Rationale$_1$]** : Grover bought 3 boxes of face masks. Each box has 20 face masks. He bought the 3 boxes for 15 dollars. He plans to sell them for 0.50 dollars each. So his total profit will be (3 * 20 * 0.50) - 15 = 15 dollars. The answer is 15.
**[Answer$_1$]** : 15. ✓

--------
**$A_1 \neq A_0$, Continue Refine** (Reorder Conditions)
**[Refined Problem$_2$]** : Grover bought 3 boxes of face masks. Each box has 20 face masks. Grover bought the 3 boxes for $15. He plans to sell the face masks for $0.50 each. How much will be his total profit?
**[Rationale$_2$]** : Grover bought 3 boxes of face masks. Each box has 20 face masks. So he bought 3 * 20 = 60 face masks in total. He bought the 3 boxes for 15 dollars. He plans to sell the face masks for 0.50 dollars each. So he will sell the 60 face masks for 60 * 0.50 = 30 dollars. His total profit will be 30 - 15 = 15 dollars.
**[Answer$_2$]** : 15. ✓

--------
**$A_2 == A_1$, Return $A_2$**

Figure 6: A case of Self-Polish on GSM-IC with Chain-of-Thought. The case is with Text-davinci-003. The irrelevant information "Grover's neighbor made a salary of $10 last year." is removed. In the second iteration, the order of the condition "Each box has 20 face masks." is moved forward and the model can calculate the total number of masks more easily when performing reasoning.

Additionally, summarizing local conditions into new ones can effectively simplify complex problems, enabling the model to handle them with greater ease. This is demonstrated in the first iteration of Figure 7 and the second iteration of Figure 8. Furthermore, the second iteration in Figure 7 highlights how our approach can explicitly and precisely define the problem in a formal manner. Specifically, in the Refined Problem$_2$ of Figure 7, the model accurately identifies the two teams as "Team A" and "Team B" instead of referring to them as "one team" and "the other team", and then it is able to clearly specify the exact question to be asked. This significantly reduces the model's burden of understanding during the reasoning process, enhancing its overall performance.

## 6 Conclusion

This paper focuses on a previously neglected aspect, namely the optimization of problem formulation, within the context of enhancing multi-step

reasoning in large language models. We present a novel prompting method called Self-Polish which progressively refines the given reasoning problems to facilitate model comprehension and processing. It demonstrates impressive effectiveness, robustness, and reliability in various benchmarks across different models, and can seamlessly integrate with other state-of-the-art methods. We hope it could motivate future research in this field.

## Limitations

Despite the significant enhancement in the reasoning performance achieved by our approach, this work still has limitations. Firstly, our criterion for convergence is based on obtaining two identical answers rather than assessing whether the problem itself has been sufficiently optimized. Future work could involve designing methods that enable the model to autonomously determine whether a problem has reached its optimal form. Secondly, we have explored two approaches to automatically construct problem-refining prompts (i.e., Auto-Sp and Complex-SP). However, in the future, it would be beneficial to incorporate more techniques for automatically generating instructions or selecting demonstrations. Thirdly, although our designed patterns for problem refining have proven highly effective, they do not encompass all possible scenarios in the real world. In the future, it is conceivable to incorporate additional patterns to further expand the scope of applicability.

## Acknowledgements

The authors wish to thank the anonymous reviewers for their helpful comments. This work was partially funded by National Natural Science Foundation of China (No.62206057,61976056,62076069), Shanghai Rising-Star Program (23QA1400200), Natural Science Foundation of Shanghai (23ZR1403500), Program of Shanghai Academic Research Leader under grant 22XD1401100.

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

# Appendix

## A   Disscussion of More Related Work

Recent research has unveiled an unpredictable phenomenon known as emergent abilities, which manifest exclusively in larger language models while eluding their smaller counterparts (Schaeffer et al., 2023). In-context learning, instruction following, and multi-step reasoning are three emergent abilities that we focus on. We have discussed the multi-step reasoning in Section 2 and we will discuss

another two. We also compare our method with the LtM detailedly here.

**In-context learning.**   It is demonstrated that a large language model can learn patterns from a few input-output examples in the context (input) to perform the task for an unseen inference-time example (Brown et al., 2020; Chowdhery et al., 2022), and such ability is referred to as in-context learning (ICL). Recent studies have further highlighted the impressive performance of ICL in reasoning tasks (Wei et al., 2022b; Fu et al., 2022; Zhou et al., 2022a). In our research, we capitalize on this capability to generate new formulations of problems by injecting rephrasing patterns to the demonstrations.

**Instruction following.**   LLMs can learn to perform unseen tasks solely through the comprehension of task-specific natual language instructions (Sanh et al., 2022; Wei et al., 2022a; Chung et al., 2022; Ouyang et al., 2022). There is also work showing that combining instructions with in-context learning can provide further benefits and that few-shot demonstrations can be viewed as a special kind of instruction that arouses the implicit ability in LLMs (Chung et al., 2022; Zhou et al., 2022b; Qiao et al., 2022).

**Compaison with LtM.**   The work that is most similar to ours may be Least-to-Most (LtM) which decomposes the original problem into a series of sub-problems that need to be solved sequentially (Zhou et al., 2022a). However, LtM is an variant of CoT, and there are differences in motivation and operation process between LtM and SP. Firstly, LtM is an answer/reasoning side approach that emphasizes the decomposition of a complex problem into sub-problems, while we emphasize refining the original problem to make it more understandable. Secondly, in LtM, sub-problems are solved sequentially, requiring the answer of the previous sub-problem to tackle the next one, which can lead to fragility in the reasoning chain. In contrast, our method allows for the combination of local related conditions to form new conditions parallelly.

## B   The Algorithm of Self-Polish

See Algorithm 1 for the overall framework of Self-Polish.

## C   Implementation Details

We set the maximum iterating count to $T = 2$. Note that the bigger maximum iteration count $T$

**Algorithm 1:** Self-Polish Prompting

**Input:** language model $\mathcal{G}$, problem set $\mathcal{S}$, prompt $\mathcal{P}_{refine}$ of the problem side refining method, prompt $\mathcal{P}_{answer}$ of the answer/reasoning side method, max iteration number $T$, answer selection strategy $\mathcal{Z}$.

1 **for** each problem $s$ in $\mathcal{S}$ **do**
2    answer_list = [ ];
3    $t = 0$;
4    **Procedure** GENERATE ANSWER TO ORIGINAL PROBLEM
5    rationale$_t$, ans$_t = \mathcal{G}(\mathcal{P}_{answer} \oplus s)$;
6    answer_list.append(ans$_t$);
7    $t = t + 1$;
8    **Procedure** ITERATE PROBLEM REFINEMENT AND ANSWER
9    $s = \mathcal{G}(\mathcal{P}_{refine} \oplus s)$;
10    rationale$_t$, ans$_t = \mathcal{G}(\mathcal{P}_{answer} \oplus s)$;
11    **if** ans$_t$ == ans$_{t-1}$ **then**
12      Return ans$_t$.
13    **else if** $t > T$ **then**
14      Return $\mathcal{Z}$(answer_list).
15    **else**
16      answer_list.append(ans$_t$);
17      $t = t + 1$;
18    **end**
19 **end**

may lead to better performance, but here we set it to 2 to achieve a trade-off between computational efficiency and effectiveness.

When combining with other reasoning-side methods (i.e., CoT, LtM, Complex-CoT and Auto-CoT) on MultiArith and SVAMP, we set the answer selection strategy as "selecting the answer to the original problem" because this dataset is relatively easy for these prompting methods. Actually, in cases where it is not necessary, rewriting easy problems may result in the loss of critical information or altering the semantics of the original problem. In other settings, we set the answer selection strategy as "selecting the answer to the last problem".

## D  More Cases and Examples

Here we list more cases of Self-Polish in Figure 7 and Figure 8. We also list the failure case of excessive problem refining in Figure 9

**[Original Problem$_0$]** : Two track teams are competing against each other in a 4 by 400 meter relay; a race where each competing team has four members that each run 400 meters, or one lap, around a standard track. One of the two teams is very well-rounded and each of their members will run their 400 meter leg in precisely 55 seconds. The other team is less well-rounded; their first runner will run their 400 meter leg in 60 seconds then each subsequent runner will be 3 seconds faster than the previous runner. Using this information, how many seconds will the faster team win by?

**[Rationale$_0$]** : The well-rounded team will run their 400 meter legs in 55 seconds each, for a total time of 220 seconds. The less well-rounded team will run their 400 meter legs in 60, 57, 54, and 51 seconds respectively, for a total time of 222 seconds. The faster team will win by 220 - 222 = -2 seconds. The answer is -2 seconds.

**[Answer$_0$]** : -2. ✗

- - - - - - - - - - - - - - - - - - - - - - - - - - -

**Begin, Problem Refine** (Summary Local Conditions)

**[Refined Problem$_1$]** : Two track teams are competing against each other in a 4 by 400 meter relay. One team has four members that each run 400 meters in 55 seconds, while the other team has four members that run 400 meters in 60, 57, 54, and 51 seconds respectively. How many seconds will the faster team win by?

**[Rationale$_1$]** : The faster team has four members that run 400 meters in 55, 60, 57, and 54 seconds respectively. The slower team has four members that run 400 meters in 51 seconds each. So the faster team will win by 55 - 51 = 4 seconds. The answer is 4.

**[Answer$_1$]** : 4. ✗

- - - - - - - - - - - - - - - - - - - - - - - - - - -

**A$_1$ != A$_0$, Continue Refine** (Specify the Problem Formally)

**[Refined Problem$_2$]** : Two track teams are competing against each other in a 4 by 400 meter relay. Team A has four members that each run 400 meters in 55 seconds, while Team B has four members that run 400 meters in 60, 57, 54, and 51 seconds respectively. How many seconds faster will Team A finish the relay compared to Team B?

**[Rationale$_2$]** : Team A has four members that each run 400 meters in 55 seconds. Team B has four members that run 400 meters in 60, 57, 54, and 51 seconds respectively. So the total time for Team A is 4 * 55 = 220 seconds, and the total time for Team B is 60 + 57 + 54 + 51 = 222 seconds. Team A will finish the relay 2 seconds faster than Team B. The answer is 2.

**[Answer$_2$]** : 2. ✓

- - - - - - - - - - - - - - - - - - - - - - - - - - -

**A$_2$ != A$_1$, Exceed Max Iterating Times, Return A$_2$**

Figure 7: A case of Self-Polish on GSM8K with Chain-of-Thought. In the first iteration, some irrelevant information is removed and the average time each member of the second team consumes is clarified. In the second iteration, the model accurately identifies the two teams as "Team A" and "Team B" instead of referring to them as "one team" and "the other team", and it explicitly states what the question to be asked is, reducing the burden of understanding on the model during the reasoning process.

## E  Sensitivity to Number and Order of Demonstrations

As widely recognized, in-context learning is highly sensitive to the number and order of demonstrations within the prompt (Min et al., 2022; Lu et al., 2022; Liu et al., 2022). In this regard, we investigate whether our problem-refining process is sensitive to these variables via experiments on GSM8K with Text-davinci-003. We randomly select 200 examples from the test set. For a specific shot number, we randomly select five sets of demonstrations. For each set of demonstrations, we obtain performance results in five different orders. We observed that in the standard manner, increasing the number of demonstrations leads to improved performance. However, in the CoT manner, the performance con-

[**Original Problem₀**] : Charlie wants to sell beeswax candles. For every pound of beeswax, he can make 10 tapered candles. One pound of beeswax and the wicks cost $10.00 in supplies. If he sells each candle for $2.00 each, what is his net profit if he makes and sells 20 candles?

[**Rationale₀**] : **Charlie needs 1 pound of beeswax and wicks to make 10 candles, which costs $10. He sells each candle for $2.00. So for 20 candles, he will make a profit of 20 * 2 = 40 dollars. His net profit is 40 - 10 = 30 dollars. The answer is 30.**

[**Answer₀**] : 30. ❌

- - - - - - - - - - - - - - - - - - - - - - - - - - - - - - - -

**Begin**, Problem Refine

[**Refined Problem₁**] : Charlie wants to sell beeswax candles. He needs 1 pound of beeswax and wicks to make 10 tapered candles, which cost him $10.00 in supplies. He sells each candle for $2.00. How much is his net profit if he makes and sells 20 candles?

[**Rationale₁**] : **Charlie needs 1 pound of beeswax and wicks to make 10 tapered candles, which cost him $10.00 in supplies. He sells each candle for $2.00. So he makes a profit of $2.00 per candle. If he makes and sells 20 candles, his net profit is 20 * 2 = 40 dollars. The answer is 40 dollars.**

[**Answer₁**] : 40. ❌

- - - - - - - - - - - - - - - - - - - - - - - - - - - - - - - -

**A₁ != A₀, Continue Refine**

[**Refined Problem₁**] : Charlie wants to sell beeswax candles. He needs 2 pounds of beeswax and wicks to make 20 tapered candles, which cost him $20.00 in supplies. He sells each candle for $2.00. How much is his net profit if he makes and sells 20 candles?

[**Rationale₂**] : **Charlie spent $20.00 in supplies to make 20 tapered candles. He sells each candle for $2.00. So he makes 20 * 2 = 40 dollars in sales. His net profit is 40 - 20 = 20 dollars. The answer is 20.**

[**Answer₂**] : 20. ✔

- - - - - - - - - - - - - - - - - - - - - - - - - - - - - - - -

**A₂ != A₁, Exceed Max Iterating Times, Return A₂**

Figure 8: A case of Self-Polish on GSM8K with Chain-of-Thought. In the first iteration, the model translates "every" to "1". In the second iteration, the model summarizes relevant conditions and specifies the total cost for Charlie.

---

[**Original Problem₀**] : A teacher had 38 worksheets to grade. If she graded 4, but then another 15 were turned in, how many worksheets would she have to grade?

[**Rationale₀**] : **The teacher had 38 worksheets to grade originally. She graded 4 of them, and then 15 more were turned in. So she had 38 - 4 = 34 worksheets to grade, and then 15 more were added. Now she has 34 + 15 = 49 worksheets to grade. The answer is 49.**

[**Answer₀**] : 49. ✔

- - - - - - - - - - - - - - - - - - - - - - - - - - - - - - - -

**Begin**, Problem Refine

[**Refined Problem₁**] : A teacher had 38 worksheets to grade. After grading 4 worksheets, 15 more were turned in. How many worksheets does the teacher now have to grade?

[**Rationale₁**] : **The teacher had 38 worksheets to grade originally. After grading 4 worksheets, 15 more were turned in. So the teacher now has 38 + 15 = 53 worksheets to grade. The answer is 53.**

[**Answer₁**] : 53. ❌

- - - - - - - - - - - - - - - - - - - - - - - - - - - - - - - -

**A₁ != A₀, Continue Refine**

[**Refined Problem₂**] A teacher had 38 worksheets to grade. After grading 4 worksheets, 15 additional worksheets were turned in. How many worksheets does the teacher now have to grade in total?

[**Rationale₂**] : **The teacher had 38 worksheets to grade originally. After grading 4 worksheets, 15 additional worksheets were turned in. So the teacher now has 38 + 15 = 53 worksheets to grade in total. The answer is 53.**

[**Answer₂**] : 53. ❌

- - - - - - - - - - - - - - - - - - - - - - - - - - - - - - - -

**A₂ == A₁, Return A₂**

Figure 9: A failure case of Self-Polish on MultiArith with Chain-of-Thought. The semantics of the second generated problem have deviated from the semantics of the original problem, and the model cannot answer correctly.

verges when the number of shots is equal to 5, demonstrating impressive sample efficiency. Additionally, in the standard manner, our method is not sensitive to the order of demonstrations while it is

---

| Method | Shots | Mean | Order Deviation |
|--------|-------|------|-----------------|
| Std+SP | 2 | 21.3 | 0.8 |
|        | 3 | 22.5 | 0.6 |
|        | 4 | 24.1 | 0.9 |
|        | 5 | 25.0 | 1.5 |
|        | 6 | 26.3 | 0.9 |
| CoT+SP | 2 | 59.0 | 2.0 |
|        | 3 | 61.3 | 1.5 |
|        | 4 | 61.5 | 1.1 |
|        | 5 | 62.1 | 1.8 |
|        | 6 | 61.3 | 1.6 |

Table 4: Sensitivity to the number and order of problem-refining demonstrations. Mean represents the average performance for the current number of shots, while the order deviation represents the average standard deviation introduced by different demonstration orders. The results are with Text-davinci-003. In the problem side, we leverage the in-context SP.

| Method | MATH |
|--------|------|
| No Refinement | 21 |
| Zero-shot SP | 23.5 |
| In-context SP | 24.5 |

Table 5: More results on MATH dataset, using Chain-of-Thought as the answer-side method

highly sensitive to the order of demonstrations in the CoT manner.

## F   Prompts of Self-Polish

The in-context Self-Polish prompt for AQuA and MathQA is in Table 6. The Auto-SP prompt for AQuA and MathQA is in Table 7 and Table 8. The Complex-SP prompt for AQuA and MathQA is in Table 9 and Table 10.

The in-context Self-Polish prompt for GSM8K, SVAMP and MultiArith is in Table 11. The Auto-SP prompt for GSM8K, SVAMP and MultiArith is in Table 12 and Table 13. The Complex-SP prompt for GSM8K, SVAMP and MultiArith is in Table 14 and Table 15.

## G   More results on MATH dataset

As Table 5 shows, we also conducted Self-Polish methods on the MATH dataset (Hendrycks et al., 2021). Our approach demonstrated promising results. Specifically, we randomly selected 200 samples for testing, and use the Chain-of-Thought as the answer-side method.

Please rewrite new versions of the original mathematical question (including the context and the final question) to be more understandable and easy to answer. Don't omit any useful information, especially the numbers.

**Original Question**: Krishan and Nandan jointly started a business. Krishan invested six times as Nandan did and invested his money for double time as compared to Nandan. Nandan earned Rs. 6000. If the gain is proportional to the money invested and the time for which the money is invested then the total gain was? Answer Choices: (A) Rs.78000 (B) Rs.48000 (C) Rs.6000 (D) Rs.82000 (E) Rs.32000
**New Question**: Krishan and Nandan teamed up to start a business together. Krishan invested 12 times more money than Nandan did. Nandan's earnings from the business were Rs. 6000. If the gain is directly proportional to both the amount of money invested and the time period, what was the total gain for both of them? Answer Choices: (A) Rs.78000 (B) Rs.48000 (C) Rs.6000 (D) Rs.82000 (E) Rs.32000

**Original Question**: In a graduate physics course, 70 percent of the students are male and 30 percent of the students are married. If two-sevenths of the male students are married, what fraction of the male students is single? Answer Choices: (A) 2/7 (B) 1/3 (C) 1/2 (D) 2/3 (E) 5/7
**New Question**: In a graduate physics course, 7/10 of the students are male and 3/10 of the students are married. If 2/7 of the male students are married, what fraction of the male students is single? Answer Choices: (A) 2/7 (B) 1/3 (C) 1/2 (D) 2/3 (E) 5/7

**Original Question**: A train 500m long can cross an electric pole in 20 sec and then find the speed of the train? Answer Choices: (A) 95 Kmph (B) 90 Kmph (C) 92 Kmph (D) 95 Kmph (E) 98 Kmph
**New Question**: A train, which is 500 meters long, takes 20 seconds to pass by an electric pole. What is the speed of the train? Represent the answer in units from answer options. Answer Choices: (A) 95 Kmph (B) 90 Kmph (C) 92 Kmph (D) 95 Kmph (E) 98 Kmph

**Original Question**: A train covers a distance of 10km in 10 min. If it takes 6 sec to pass a telegraph post, then the length of the train is? Answer Choices: (A) 50m (B) 60m (C) 100m (D) 90m (E) 120m
**New Question**: A train covers a distance of 10000m in 600 sec. If it takes 6 sec to pass a telegraph post, then the length of the train is? Answer Choices: (A) 50m (B) 60m (C) 100m (D) 90m (E) 120m

**Original Question**: How many different subsets of the set {0, 1, 2, 3, 4} do not contain 0? Answer Choices: (A) 16 (B) 27 (C) 31 (D) 32 (E) 64
**New Question**: How many different subsets of the set { 1, 2, 3, 4} ? Answer Choices: (A) 16 (B) 27 (C) 31 (D) 32 (E) 64

**Original Question**: A class has 6 boys and x girls. Average score of boys and girls is 50 and 60 respectively. the average of the whole class is 55, what is the value of x? Answer Choices: (A) 5 (B) 6 (C) 10 (D) 12 (E) 15
**New Question**: In a class, there are 6 boys and an unknown number of girls. The average score of the boys is 50, while the average score of the girls is 60. The overall average score of the boys and girls is 55. How many girls are there in the class? Answer Choices: (A) 5 (B) 6 (C) 10 (D) 12 (E) 15

Table 6: In-context SP prompt for AQuA and MathQA.

Please rewrite new versions of the original mathematical question (including the context and the final question) to be more understandable and easy to answer. Don't omit any useful information, especially the numbers.

**Original Question**: A and B can together finish a work in 40days. They worked together for 10days and then B left. After another 12days, A finished the remaining work. In how many days A alone can finish the job? Answer Choices: (A) 10 (B) 25 (C) 60 (D) 16 (E) 20
**New Question**: A and B can together finish a work in 40 days. They worked together for 10 days and then B left, and the remaining work is 3/4 of the original one. After another 12 days, A finished the remaining work alone. In how many days A alone can finish the whole job? Answer Choices: (A) 10 (B) 25 (C) 60 (D) 16 (E) 20

**Original Question**: A man buys an article and sells it at a profit of 20%. If he had bought it at 20% less and sold it for Rs.75 less, he could have gained 25%. What is the cost price? Answer Choices: (A) 388 (B) 375 (C) 288 (D) 266 (E) 269
**New Question**: A man buys an article at the price of x and sold it at the price of 1.2x, if he had bought it at a 20% discount which is 0.8x and sold it for Rs.75 less than 1.2x, he would have gained 25% of 0.8x. What was the original price of the article before any discounts or markups? Answer Choices: (A) 388 (B) 375 (C) 288 (D) 266 (E) 269

**Original Question**: The numbers of students speaking English and Hindi are in the ratio of 4 : 5. If the number of students speaking English increased by 35% and that speaking Hindi increased by 20%, what would be the new respective ratio? Answer Choices: (A) 19 : 20 (B) 7 : 8 (C) 8 : 9 (D) Cannot be determined (E) None of these
**New Question**: The number of students speaking English is 400 and increased by 35%. The number of students speaking Hindi is 500 and increased by 20%, what would be the respective ratio? what is the new ratio of students speaking English and Hindi? Answer Choices: (A) 19 : 20 (B) 7 : 8 (C) 8 : 9 (D) Cannot be determined (E) None of these

**Original Question**: A rectangular field has area equal to 150 sq m and perimeter 50 m. Its length and breadth must be? Answer Choices: (A) 10 (B) 88 (C) 66 (D) 65 (E) 22
**New Question**: Let l and b be the length and the breadth of the rectangular. The area of a rectangular field is 150 square meters: l*b = 50, and its perimeter is 50 meters: 2l + 2b =50. What are the breadth of the field? Answer Choices: (A) 10 (B) 88 (C) 66 (D) 65 (E) 22

**Original Question**: The ratio of two numbers is 3:4 and their sum is 14. The greater of the two numbers is? Answer Choices: (A) 12 (B) 14 (C) 16 (D) 8 (E) 19
**New Question**: There are two number a and b. The sum of a and b is 14, and the ratio of a and b 3:4. What is b? Answer Choices: (A) 12 (B) 14 (C) 16 (D) 8 (E) 19

**Original Question**: A and B invests Rs.6000 and Rs.8000 in a business. After 6 months, A withdraws half of his capital and B withdraws one-fourth of his capital. In what ratio should they share the profits at the end of the year? Answer Choices: (A) 13:15 (B) 9:13 (C) 9:11 (D) 13:14 (E) 9:14
**New Question**: A and B invested Rs.6000 and Rs.8000 respectively in a business. After 6 months,A withdraws half of his investment and B withdraws 1/4 of his investment. What is the ratio of their remaining investment? Answer Choices: (A) 13:15 (B) 9:13 (C) 9:11 (D) 13:14 (E) 9:14

Table 7: Auto-SP prompt for AQuA and MathQA.

**Original Question**: A train 640 meters long is running with a speed of 64 kmph. The time taken by it to cross a tunnel 140 meters long is? Answer Choices: (A) 44 sec (B) 49 sec (C) 48 sec (D) 16 sec (E) 17 sec

**New Question**: A train is running with a speed of 64kmph. The length of train is 640 meters and there is a tunnel 140 meters long. The time taken by the train to cross tunnel is? Answer Choices: (A) 44 sec (B) 49 sec (C) 48 sec (D) 16 sec (E) 17 sec

**Original Question**: There are 15 boys and 10 girls in a class. If three students are selected at random, in how many ways that 1 girl and 2 boys are selected ? Answer Choices: (A) 950 (B) 1050 (C) 2150 (D) 2050 (E) 1000

**New Question**: There are 15 boys and 10 girls in a class. If three students are selected at random, how many total ways can 1 girl be chosen in 10 girls and 2 boys be chosen in 15 boys? Answer Choices: (A) 950 (B) 1050 (C) 2150 (D) 2050 (E) 1000

Table 8: Continuation of Auto-SP prompt for AQuA and MathQA.

Please rewrite new versions of the original mathematical question (including the context and the final question) to be more understandable and easy to answer. Don't omit any useful information, especially the numbers.

**Original Question**: There were 35 students in a hostel. Due to the admission of 7 new students the expenses of the mess were increased by rs .84 per day while the average expenditure per head diminished by re 1. What was the original expenditure of the mess? Answer Choices: (A) rs 450 (B) rs 920 (C) rs 550 (D) rs . 630 (E) none of these
**New Question**: In a hostel, there were initially 35 students, and then 7 new students were admitted. While the average expenditure per student decreased by Re. 1, the daily expenses of the new mess increased by Rs. 84. What was the original daily expenditure of the mess? Answer Choices: (A) rs 450 (B) rs 920 (C) rs 550 (D) rs . 630 (E) none of these

**Original Question**: A train 200m long passes a man , running at 5km / hr in the same direction in which the train is going, in 10 seconds. The speed of the train is? Answer Choices: (A) 28 (B) 50 (C) 77 (D) 22 (E) 12
**New Question**: A train, which is 200 meters long, passes a man running at 5 kilometers per hour in the same direction as the train in 10 seconds. What is the speed of the train? Answer Choices: (A) 28 (B) 50 (C) 77 (D) 22 (E) 12

**Original Question**: Solution x contains 20 % of material a and 80 % of material b . solution y contains 30 % of material a and 70 % of material b . a mixture of both these solutions contains 22 % of material a in the final product . How much solution x is present in the mixture ? Answer Choices: (A) 40 % (B) 60 % (C) 80 % (D) 100 % (E) 110 %
**New Question**: A mixture of solution x and y contains 22 % of material a. Solution x contains 20 % of material a and 80 % of material b while solution y contains 30 % of material a and 70 % of material b. What percentage of solution x is present in the mixture? Answer Choices: (A) 40 % (B) 60 % (C) 80 % (D) 100 % (E) 110 %

**Original Question**: A trader sells 40 metres of cloth for rs.8200 at a profit of rs.35 per metre of cloth. How much profit will the trader earn on 40 metres of cloth ? Answer Choices: (A) rs.950 (B) rs . 1500 (C) rs . 1000 (D) rs . 1400 (E) none of these
**New Question**: A trader sells 40 meters of cloth and makes a profit of Rs. 35 per meter of cloth. How much profit does the trader make from selling 40 meters of cloth? Answer Choices: (A) rs . 950 (B) rs . 1500 (C) rs . 1000 (D) rs . 1400 (E) none of these

**Original Question**: If x < y < z and y - x > 5 , where x is an even integer and y and z are odd integers , what is the least possible value s of z - x ? Answer Choices: (A) 6 (B) 7 (C) 8 (D) 9 (E) 10
**New Question**: If x is an even integer, y and z are odd integers, and y is greater than x by more than 5, and z is greater than y. What is the smallest possible difference between z and x? Answer Choices: (A) 6 (B) 7 (C) 8 (D) 9 (E) 10

**Original Question**: What is the difference between the c.i. on rs . 6000 for 1 1/2 years at 4 % per annum compounded yearly and half-yearly? Answer Choices: (A) s.2.04 (B) s.2.08 (C) s.2.02 (D) s.2.83 (E) s.2.45
**New Question**: What is the difference in the compound interest earned on Rs. 6000 for 1.5 years at 4% per annum when compounded yearly and when compounded half-yearly? Answer Choices: (A) s.2.04 (B) s.2.08 (C) s.2.02 (D) s.2.83 (E) s.2.45

Table 9: Complex-SP prompt for AQuA and MathQA.

**Original Question**: The average weight of a, b and c is 45 kg. If the average weight of a and b be 40 kg and that of b and c be 45 kg , then the weight of b is? Answer Choices: (A) 31 kg (B) 32 kg (C) 33 kg (D) 35 kg (E) none of these

**New Question**: The average weight of a, b and c is 45 kg, which means the total weight of a, b and c is 135 kg. If the average weight of a and b is 40 kg, which means the total weight of a and b is 80kg, so the weight of c is 45kg. The average weight of b and c is 45 kg which means the total weight of b and c is 90kg. What is the weight of b? Answer Choices: (A) 31 kg (B) 32 kg (C) 33 kg (D) 35 kg (E) none of these

**Original Question**: The compound and the simple interests on a certain sum at the same rate of interest for two years are rs.11730 and rs.10200 respectively . The sum is? Answer Choices: (A) rs.17037 (B) rs.17000 (C) rs.17276 (D) rs.170287 (E) rs.171881

**New Question**: A sum of money earns compound interest and simple interest at the same rate for two years. The compound interest is Rs.11730 and the simple interest is Rs.10200. What is the sum of money? Answer Choices: (A) rs.17037 (B) rs.17000 (C) rs.17276 (D) rs.170287 (E) rs.171881

Table 10: Continuation of Complex-SP prompt for AQuA and MathQA.

Please rewrite new versions of the original mathematical question (including the context and the final question) to be more understandable and easy to answer. Don't omit any useful information, especially the numbers.

**Original Question**: Each bird eats 12 beetles per day, each snake eats 3 birds per day, and each jaguar eats 5 snakes per day. If there are 6 jaguars in a forest, how many beetles are eaten each day?
**New Question**: In a forest, there are 6 jaguars that each eat 5 snakes per day. Each snake eats 3 birds per day, and each bird eats 12 beetles per day. How many beetles are eaten each day by the jaguars?

**Original Question**: Albert is wondering how much pizza he can eat in one day. He buys 2 large pizzas and 2 small pizzas. A large pizza has 16 slices and a small pizza has 8 slices. If he eats it all, how many pieces does he eat that day?
**New Question**: Albert has purchased 2 large pizzas and 2 small pizzas and is wondering how many slices he can eat in one day. Each large pizza has 16 slices and each small pizza has 8 slices. If Albert eats all of the pizza, how many slices will he have eaten in one day?

**Original Question**: In a truck, there are 26 pink hard hats, 15 green hard hats, and 24 yellow hard hats. If Carl takes away 4 pink hard hats, and John takes away 6 pink hard hats and twice as many green hard hats as the number of pink hard hats that he removed, then calculate the total number of hard hats that remained in the truck.
**New Question**: In a truck, there are 26 pink hard hats, 15 green hard hats, and 24 yellow hard hats. Carl takes away 4 pink hard hats and John takes away 6 pink hard hats and 12 green hard hats. How many hard hats remain in the truck?

**Original Question**: Jasper will serve charcuterie at his dinner party. He buys 2 pounds of cheddar cheese for $10, a pound of cream cheese that cost half the price of the cheddar cheese, and a pack of cold cuts that cost twice the price of the cheddar cheese. How much does he spend on the ingredients?
**New Question**: Jasper is hosting a dinner party and wants to serve charcuterie. He buys 2 pounds of cheddar cheese for $10, 1 pound of cream cheese for $5, and a pack of cold cuts for $20. How much does he spend on the ingredients for the charcuterie?

**Original Question**: Tomas ate 1.5 pounds of chocolate fudge last week. Katya ate half a pound of peanut butter fudge, while Boris ate 2 pounds of fudge. How many ounces of fudge did the Tomas, Katya and Boris eat in total?
**New Question**: Tomas ate 24 ounces of chocolate fudge last week. Katya ate 8 ounces of peanut butter fudge, while Boris ate 32 ounces of fudge. How many ounces of fudge did the Tomas, Katya and Boris eat in total?

**Original Question**: Tomas ate 24 ounces of chocolate fudge last week. Katya ate 8 ounces of peanut butter fudge, while Boris ate 32 ounces of fudge. How many ounces of fudge did the Tomas, Katya and Boris eat in total?
**New Question**: Tomas ate 24 ounces of fudge last week. Katya ate 8 ounces of fudge, while Boris ate 32 ounces of fudge. How many ounces of fudge did the Tomas, Katya and Boris eat in total?

Table 11: In-context SP prompt for GSM8K, SVAMP and MultiArith.

Please rewrite new versions of the original mathematical question (including the context and the final question) to be more understandable and easy to answer. Don't omit any useful information, especially the numbers.

**Original Question**: Monica is a teacher. She has 6 classes per day. The first class has 20 students. The second and third classes have 25 students. Her fourth class has half as many as her first class. Her fifth and sixth classes have 28 students. How many students does Monica see each day?
**New Question**: Monica is a teacher with 6 classes per day. Her first class has 20 students, her second and third classes have 25 students, and her fourth class has 10 students. Her fifth and sixth classes have 28 students. How many students does Monica see each day in all of her classes?

**Original Question**: Emily went to the store and bought art supplies for $20 and 2 skirts that cost the same amount of money. She spent a total of $50. How much did Emily pay for each of the skirts?
**New Question**: Emily went to the store and bought art supplies for $20 and 2 skirts for a total of $50. How much did Emily pay for each of the skirts?

**Original Question**: John's neighbor tells him to walk his dog for 1 hour each day for a total of $10. He does this for April, save for the 4 Sundays in April. He later spent $50 on books and gave his sister Kaylee the same amount. How much money did John have left?
**New Question**: John's neighbor tells him to walk his dog for April (30 days excluding 4 Sundays) for a total of $10 each day. He later spent $50 on books and gave his sister Kaylee the same amount. How much money did John have left after these expenses?

**Original Question**: Three years ago, Bethany was twice the age of her younger sister. In 5 years, her younger sister will be 16. How old is Bethany now?
**New Question**: Three years ago, Bethany was twice the age of her younger sister, who is currently 11 years old. How old is Bethany now?

**Original Question**: At the bookstore, Sarah bought 6 paperback books and 4 hardback books. Her brother bought one-third as many paperback books as Sarah bought, and two times the number of hardback books that she bought. How many books did her brother buy in total?
**New Question**: At the bookstore, Sarah bought 6 paperback books and 4 hardback books. Her brother bought 2 paperback books and 8 hardback books. How many books did her brother buy in total?

**Original Question**: Sandra had 2 different bags of candy. Each of her bags had 6 pieces of candy left. Her brother, Roger, also had 2 bags of candy. One of his bags of candy had 11 pieces left and the other had 3 pieces left. How much more candy did Roger have?
**New Question**: Sandra had 2 bags of candy, each with 6 pieces left. Her brother, Roger, had 2 bags of candy, one with 11 pieces left and the other with 3 pieces left. How many more pieces of candy did Roger have than Sandra?

**Original Question**: Joan wants to visit her family who live 480 miles away. If she drives at a rate of 60 mph and takes a lunch break taking 30 minutes, and 2 bathroom breaks taking 15 minutes each, how many hours did it take her to get there?
**New Question**: Joan wants to visit her family who live 480 miles away. If she drives at a rate of 60 mph and takes a lunch break of 30 minutes, and 2 bathroom breaks of 15 minutes each, how many hours(60 minutes = 1 hour) does it take her to get there?

Table 12: Auto-SP prompt for GSM8K, SVAMP and MultiArith.

**Original Question**: James gets a fleet of gas transportation vans. He gets 6 vans. 2 of them are 8000 gallons. 1 of them is 30% less than that. The remaining trucks are 50% larger than the 2 trucks. How many gallons can he transport?

**New Question**: James has acquired a fleet of gas transportation vans. He has 6 vans in total. 2 of the vans have a capacity of 8000 gallons, while the other van has a capacity of 5600 gallons (30% less than the first two vans). The remaining 3 vans have a capacity of 12000 gallons (50% larger than the first two vans). What is the total capacity of the fleet in gallons?

Table 13: Continuation of Auto-SP prompt for GSM8K, SVAMP and MultiArith.

Please rewrite new versions of the original mathematical question (including the context and the final question) to be more understandable and easy to answer. Don't omit any useful information, especially the numbers.

**Original Question**: Angelo and Melanie want to plan how many hours over the next week they should study together for their test next week. They have 2 chapters of their textbook to study and 4 worksheets to memorize. They figure out that they should dedicate 3 hours to each chapter of their textbook and 1.5 hours for each worksheet. If they plan to study no more than 4 hours each day, how many days should they plan to study total over the next week if they take a 10-minute break every hour, include 3 10-minute snack breaks each day, and 30 minutes for lunch each day?
**New Question**: Angelo and Melanie want to plan how many hours over the next week they should study together for their test next week. They have 2 chapters of their textbook to study and they decide to dedicate 3 hours to each chapter. They also have 4 worksheets to memorize, and they decide to dedicate 1.5 hours for each worksheet. Taking into account 10-minute breaks every hour, if they plan to study no more than 4 hours each day including 3 10-minute snack breaks each day, and 30 minutes for lunch each day, how many days should they plan to study total over the next week?

**Original Question**: Mark's basketball team scores 25 2 pointers, 8 3 pointers and 10 free throws. Their opponents score double the 2 pointers but half the 3 pointers and free throws. What's the total number of points scored by both teams added together?
**New Question**: Mark's basketball team scores 25 2 pointers, 8 3 pointers and 10 free throws. Their opponents score 50 2 pointers, 4 3 pointers and 5 free throws. Both teams score 75 2 pointers, 12 3 pointers and 15 free throws. What is the total number of points scored by both teams combined?

**Original Question**: Bella has two times as many marbles as frisbees. She also has 20 more frisbees than deck cards. If she buys 2/5 times more of each item, what would be the total number of the items she will have if she currently has 60 marbles?
**New Question**: Bella currently has 60 marbles, and she has twice as many marbles as frisbees and 20 more frisbees than deck cards. She buys 2/5 times more of each item. What would be the total number of the items she will have?

**Original Question**: A group of 4 fruit baskets contains 9 apples, 15 oranges, and 14 bananas in the first three baskets and 2 less of each fruit in the fourth basket. How many fruits are there?
**New Question**: There is a group of 4 fruit baskets. The first three baskets each contains 9 apples, 15 oranges, and 14 bananas, and 7 apples, 13 oranges, and 12 bananas in the fourth basket. How many fruits are there in total?

**Original Question**: You can buy 4 apples or 1 watermelon for the same price. You bought 36 fruits evenly split between oranges, apples and watermelons, and the price of 1 orange is $0.50. How much does 1 apple cost if your total bill was $66?
**New Question**: You bought 36 fruits, with an equal number of oranges, apples and watermelons. The price of 1 watermelon equals to 4 apples, and the price of 1 orange is $0.50. If your total bill was $66, how much does 1 apple cost?

Table 14: Complex-SP prompt for GSM8K, SVAMP and MultiArith.

**Original Question**: Susy goes to a large school with 800 students, while Sarah goes to a smaller school with only 300 students. At the start of the school year, Susy had 100 social media followers. She gained 40 new followers in the first week of the school year, half that in the second week, and half of that in the third week. Sarah only had 50 social media followers at the start of the year, but she gained 90 new followers the first week, a third of that in the second week, and a third of that in the third week. After three weeks, how many social media followers did the girl with the most total followers have?

**New Question**: At the start of the school year, Susy had 100 social media followers and Sarah had 50 social media followers. Susy gained 40 followers in the first week, 20 in the second week, and 10 in the third week. Sarah gained 90 followers in the first week, 30 in the second week, and 10 in the third week. After three weeks, how many social media followers did the girl with the most total followers have?

**Original Question**: Sam bought a dozen boxes, each with 30 highlighter pens inside, for $10 each box. He rearranged five of these boxes into packages of six highlighters each and sold them for $3 per package. He sold the rest of the highlighters separately at the rate of three pens for $2. How much profit did he make in total, in dollars?

**New Question**: Sam bought 12 boxes for $10 each, and each contains 30 highlighter pens. 1 package contains 6 highlighters. He rearranged five of these boxes into packages and sold them for $3 per package. He sold the remaining highlighters separately at the price of $2 for every three one. How much profit did Sam make in total, in dollars?

**Original Question**: In a certain school, 2/3 of the male students like to play basketball, but only 1/5 of the female students like to play basketball. What percent of the population of the school do not like to play basketball if the ratio of the male to female students is 3:2 and there are 1000 students?

**New Question**: In a certain school, there is a total of 1000 students, while 3 male students for every 2 female students. So there are 600 male students and 2/3 of the male students like to play basketball, and there are 400 female students but only 1/5 of the female students like to play basketball. What percent of the population of the school do not like to play basketball?

Table 15: Continuation of Complex-SP prompt for GSM8K, SVAMP and MultiArith.