# OpenReview forum: "Self-Polish: Enhance Reasoning in Large Language Models via Problem Refinement"
_EMNLP/2023/Conference — EMNLP 2023 Findings_

### Official Review · Reviewer_Y5Nk · 2023-08-04

**Soundness:** 3

**Excitement:**

2: Mediocre: This paper makes marginal contributions (vs non-contemporaneous work), so I would rather not see it in the conference.

**Missing References:**

@misc{press2023measuring,
      title={Measuring and Narrowing the Compositionality Gap in Language Models},
      author={Ofir Press and Muru Zhang and Sewon Min and Ludwig Schmidt and Noah A. Smith and Mike Lewis},
      year={2023},
      eprint={2210.03350},
      archivePrefix={arXiv},
      primaryClass={cs.CL}
}

**Paper Topic And Main Contributions:**

This paper focuses on the problem side of prompting tech, called self-polish, refining the format of problems to improve the performance of reasoning tasks.

Main contributions include:
1. thorough experiments about the variations of few-shot demonstrations proving the effectiveness of such methods.

**Questions For The Authors:**

1. Do you have a detailed discussion about the performance differences between all SP methods?
2. Can all SP methods be combined together?

**Reasons To Accept:**

1. a story about the prompting mechanism on the problem side, refining the problem to make it easily understood for the model. Thorough experiments are executed showing significant improvement.
2. the idea to refine the problem sounds reasonable.

**Reasons To Reject:**

1. although the results are promising, the way to refine the problem seems too simple as stated in the limitation part by the authors. It lies in the scope of the few-shot demonstration method, which is already widely discussed in previous papers. A deep exploration of how to refine the problem (decompose questions for example) is highly expected.
2. all sub-ideas (auto-sp for example) come from other works, which makes it look like a work combined with multiple trivial ideas.

**Reproducibility:**

4: Could mostly reproduce the results, but there may be some variation because of sample variance or minor variations in their interpretation of the protocol or method.

**Reviewer Confidence:**

4: Quite sure. I tried to check the important points carefully. It's unlikely, though conceivable, that I missed something that should affect my ratings.

---

> ### Author Rebuttal · Authors · 2023-08-29
>
> Thank you very much for your recognition of our idea to enhance LLM reasoning performance from the problem-side and your affirmation of the completeness of our work. We sincerely appreciate your feedback, and we will diligently work towards improving the manuscript. We hope we can address your concerns.
>
> 1. **Question of the simplicity of the ways to refine the problems.**
>    - Thanks for your question. We acknowledge the concerns raised about the simplicity of our approach, especially in relation to refining the problem. Specifically, the method and principles we introduced for problem refinement—namely, the emphasis on conciseness, clarity, focus, and the elimination of irrelevant information—may not fully encompass all the complexities present in real-world scenarios. **However, we would like to assert that the simplicity of our methodology is not a** **shortcoming** **but rather a strength of our work. We believe that simplicity does not equate to a lack of innovation**. In fact, often the most impactful ideas in science and technology are those that are elegantly simple, yet transformative in their implications.
>    - Our work is the **first to consider the reasoning performance from the perspective of problem refinement, providing a fresh and unique viewpoint**. While the concept might seem intuitive upon reflection, there have been no prior studies that have comprehensively approached and implemented this idea in practice. Our intent was to introduce a method that starts from the **simplest and most streamlined form**, laying a robust foundation that offers ample room for future enhancements. By presenting this minimalistic approach, we hope to inspire subsequent research in this domain.
>    - We appreciate the suggestion for a deeper exploration, and we have also incorporated the question decomposition method (LtM) you mentioned. We certainly see the potential for further sophistication in future iterations. However, for this particular work, our goal was to introduce the core concept and demonstrate its potential, ensuring that the community recognizes the value of this new perspective.
>
> 2. **Question of sub-ideas (auto-sp for example).**
>    - Thank you very much for your question. While our work indeed draws inspiration from a collection of established concepts, **the uniqueness and contribution of our research lie in a new perspective and framework for enhancing the reasoning capabilities of large language models, distinct from non-contemporaneous approaches.**
>    - Building upon this foundation, we have integrated a series of influential methods for selecting demonstrations in in-context learning, such as auto-sp or complex-sp, to **validate the potential and versatility of our idea for further enhancement**. We do not perceive this as a mere amalgamation of trivial ideas; rather, **it serves as a viable avenue for future research**, showcasing that leveraging demonstration selection methods or other automatic prompting methods can further elevate the performance of self-polish.
>
> 3. **Question of a detailed discussion about the performance differences between all SP methods.**
>    - Thank you very much for your valuable suggestion. We will incorporate this discussion accordingly in the later version of our work:
>    - Overall, the **performance of Zero-shot SP is lower than that of In-context SP**. When we employ different methods to select demonstrations (In-context SP, Auto-SP, Complex-SP) and combine them with various answer-side approaches, we find that **automated methods are not consistently superior to randomly selecting methods**, as shown in Table 2. For instance, on the AQuA dataset, when using Complex-CoT, In-context SP exhibits the best performance.
>    - Additionally, we computed the likelihood of success for each SP method across the 4 \times 5 = 20 settings mentioned in Table 2. The results indicate that **Complex-CoT performs the best (9 out of 20)**, aligning with the average performance results (last column of Table 2). This underscores the model's exceptional ability to generalize from complex to simpler scenarios during problem refinement. On the other hand, **Auto-SP (4 out of 20) falls behind In-context SP (6 out of 20)**. This discrepancy might arise from the fact that the Automatic method selects samples based on the semantic categorization of problems rather than choosing samples according to potential quality issues that problems might have, potentially rendering them less informative.
>
> 4. **Question of whether all SP methods can be combined together.**
>    - Thank you very much for your question. All the SP methods cannot be combined together because their demonstrations are selected based on different criteria.
>
> 5. **Question of the missing reference.**
>    - Thank you very much for your kind reminder. We will definitely include this reference and other recent relevant literature in the upcoming manuscript.

---

### Official Review · Reviewer_Wq3x · 2023-08-04

**Soundness:** 3

**Excitement:**

3: Ambivalent: It has merits (e.g., it reports state-of-the-art results, the idea is nice), but there are key weaknesses (e.g., it describes incremental work), and it can significantly benefit from another round of revision. However, I won't object to accepting it if my co-reviewers champion it.

**Paper Topic And Main Contributions:**

The paper proposes another strategy to solve complex problems with LLMs: the Self-Polishing strategy. According to this strategy, the problem is resolved by using first a polishing procedure that makes the problem easier and standard for the final LLMs to solve. SP can be combined with existing strategies to solve complex problems based on Chain-of-Thoughts.

**Questions For The Authors:**

1) In-context Self-Polish seems to be a very interesting strategy as Self-Polish is done according to some examples. It is not clear how these curated examples (or prompts) are used, though.
2) In-context Self Polish has curated examples. How many people have done this curation? Has the inter-annotator agreement been computed? What happens if the curation change? Is it sensitive to a particular kind of curation?

**Reasons To Accept:**

- The Self-Polishing strategy seems to be effective when used in combination with Chain-of-Thoughts.

**Reasons To Reject:**

- The real contribution of the paper is not that sound. Indeed, there is no retraining. It seems to be only prompt engineering.

- One of the most important innovative points - In-context Self Polish - is rather vague. Indeed, it requires (human) curation, but no analysis of what is the effect of this curation is done. There is no comparative analysis with other sessions of curations.

- Is it possible that the research on reasoning started only in the year 2022? In the background section, there are hardly any references to works done before the year 2022. This is extremely weird.

**Reproducibility:**

3: Could reproduce the results with some difficulty. The settings of parameters are underspecified or subjectively determined; the training/evaluation data are not widely available.

**Reviewer Confidence:**

4: Quite sure. I tried to check the important points carefully. It's unlikely, though conceivable, that I missed something that should affect my ratings.

**Typos Grammar Style And Presentation Improvements:**

Figure 1 is not particularly informative. It should be paired with the real running example of Figure 2.

---

> ### Author Rebuttal · Authors · 2023-08-29
>
> Thanks for your approval of our work and your valuable suggestions. We'll make efforts to improve the quality of the next manuscript. We hope we can address your concerns.
>
> 1. **Question about the contribution of the paper and the question regarding prompt-based methods and retraining-based methods.**
>
>    - Thank you very much for your question.
>    - **Our primary contributions**: In contrast to previous methods that focused on enhancing LLMs' reasoning performance from the answer side (e.g., CoT, self-consistency), we introduce a novel perspective of approaching the problem from the problem side. By improving the quality of the reasoning problems, we enhance LLMs' reasoning capabilities. Through extensive experiments in various settings, we demonstrate that our approach can be orthogonal to other answer-side methods, yielding superior outcomes.
>    - Question regarding prompt-based methods and retraining-based methods: **The primary focus of our research is to enhance the reasoning performance of LLMs. Many influential works, such as Chain-of-Thought (CoT), are based on prompt-engineering.** It's a simple yet effective idea that boosts reasoning performance by guiding a reasoning chain on the answer side. **Given the current landscape where many models are close-sourced, the prompt-based approach has its advantages**: strong transferability and generalization capabilities, achieving impressive results without the need for extensive training, and avoiding issues like catastrophic forgetting. Additionally, a sobering reality is that not all practitioners have access to the computational resources required for extensive LLM training, even though training might lead to improvements in problem refinement and answering.
> - In summary, our work introduces **a new perspective for enhancing the reasoning performance of** **LLMs** **and achieves promising results, potentially inspiring researchers in the field**. We believe that both prompt-based research and retraining-based research have their own advantages. **As long as they drive the advancement of LLMs and NLP, they are worthwhile endeavors.**
>
> 2. **Question about the curation of In-context Self-Polish examples and relevant analysis.**
>
>    - Thank you very much for your question and valuable suggestions.
>
>    -  First, (regarding the cost of manual curation) the method of In-context Self-Polish **indeed requires manual curation, but it's a very simple and efficient process.** We randomly select examples from the training set and manually create problem refinement demonstrations. The manual effort for crafting these demonstrations is minimal. It involves refining based on the problem-refining patterns we propose (e.g., removing irrelevant information, reordering logical structures, consolidating related conditions into new conditions, etc.). This process yields good results. We provide our prompts in Appendix F of the paper. This kind of approach utilizing manual curation is a common practice in current In-context Learning and CoT methods, such as [1]. **In fact, our primary goal is to offer a perspective of problem-side refinement that serves as inspiration for researchers, rather than expending excessive human effort on curation to explore the upper limits of method performance.**
>
>    - Secondly, the prompts used in in-context self-polish exhibit **strong and insensitive generalization capabilities**. Below are some pieces of evidence and analysis:
>
>      - Both AQuA and MathQA share the same prompts, while GSM8K, SVAMP, and MultiArith share the same prompts. The results show promising performance across these datasets.
>
>      - In Appendix E and Figure 4, we conducted ablation experiments on the GSM8K dataset, **analyzing the sensitivity of the in-context self-polish method to the number and order of demonstrations in the prompts.** The results demonstrate the method's strong generalization capability.
>
>      - **We selected five different sets of examples from the training dataset and had the same person create five distinct prompts.** We conducted experiments under the setting described in Appendix E, using 200 GSM8K test examples. The results are presented in the table below, showing minimal differences in performance due to varying prompts:
>
>        - |                  | Max  | Min  | Average | Standard deviation |
>          | ---------------- | ---- | ---- | ------- | ------------------ |
>          | Std+In-contex SP | 26   | 23.5 | 24.8    | 1.04               |
>          | CoT+In-contex SP | 63   | 59.5 | 61.2    | 1.52               |
>
>      - **We selected the same set of examples from the training dataset and had five different individuals create five distinct prompts.** We conducted experiments under the setting described in Appendix E, using 200 GSM8K test examples. The results, as presented in the table below, indicate that the impact of prompts constructed by different individuals is also minimal:
>
>        - |                  | Max  | Min  | Average | Standard deviation |
>          | ---------------- | ---- | ---- | ------- | ------------------ |
>          | Std+In-contex SP | 25.5 | 23.5 | 24.5    | 0.79               |
>          | CoT+In-contex SP | 63   | 60.5 | 61.9    | 0.96               |
>
>    - Finally, although we have demonstrated the effectiveness of our idea, there is room for improvement in reasoning performance, and it's possible to achieve even better results. Additionally, we have explored some automated methods (Complex-SP and Auto-SP), and the results are highly promising.
>
>
>
> 3. **More questions about the developing process of the curated examples for in-context self-polish.**
>    - In the original paper, **three individuals participated in the example curation process: one person was responsible for crafting the prompts, while the other two individuals carried out** **validation**. All three participants reached a consensus on the final prompts. As previously mentioned, the crafting process was straightforward, making it easy to achieve agreement.
>    - As mentioned earlier, **our method is not sensitive to different curated prompts and exhibits strong generalization capabilities.**
>
> 4. **Question about the time of referenced works.**
>    - Thank you very much for your questions and suggestions. In fact, our research focuses on the topic of reasoning with Large Language Models (LLMs), based on a paradigm similar to Chain-of-Thought. This research topic emerged due to the advancements of LLMs like ChatGPT and the growing interest in Chain-of-Thought methods, leading to a majority of citations post-2022.
>    - **Following your advice, we will include additional earlier references on reasoning in NLP as a supplementary aspect in our work like [2] and [3].**
>
>
>
> [1] Wei, Jason, et al. "Chain-of-thought prompting elicits reasoning in large language models." NeurIPS 2022
>
> [2] Ling, Wang, et al. "Program Induction by Rationale Generation : Learning to Solve and Explain Algebraic Word Problems." ACL 2017
>
> [3] Talmor, Alon, et al. "Leap-Of-Thought: Teaching Pre-Trained Models to Systematically Reason Over Implicit Knowledge." NeurIPS 2020

---

### Official Review · Reviewer_8rFo · 2023-08-12

**Typos Grammar Style And Presentation Improvements:** NA
**Soundness:** 4

**Excitement:**

3: Ambivalent: It has merits (e.g., it reports state-of-the-art results, the idea is nice), but there are key weaknesses (e.g., it describes incremental work), and it can significantly benefit from another round of revision. However, I won't object to accepting it if my co-reviewers champion it.

**Missing References:**

NA

**Paper Topic And Main Contributions:**

The paper tries to enhance the multistep reasoning capabilities of LLMs by enhancing the quality of the reasoning question. The key contribution of this paper is a systematic framework for formulating higher quality problems.

**Questions For The Authors:**

A. How is the performance of the proposed method on datasets where implicit reasoning steps are required, for example, in MATH (Hendrycks et al.) and JEEBench ( Arora et al.) ?

B. How many LLM calls does the proposed method take on average for each dataset?

C. Assuming, it takes 5 LLM calls per question for datasets like AQuA and MathQA, the performance of the proposed method according to table 2 is 50.4 and 44.8 respectively in the Complex-COT case, whereas the performance with only self-consistency as per table 3, 61.4 and 61.6 respectively. How is the proposed method then better than self-consistency in terms of computational cost?

**Reasons To Accept:**

The paper explores a novel idea of problem-side refinement. Although there has been a lot of research in the direction of finding good prompts and improving multistep reasoning of LLMs by problem decomposition, none of them has explored problem refinement as a whole.

**Reasons To Reject:**

1. The solution is novel and easy to follow, but, it might not work in problems requiring implicit reasoning steps.

2. The computational benifits of the approach are unclear.

**Reproducibility:**

5: Could easily reproduce the results.

**Reviewer Confidence:**

4: Quite sure. I tried to check the important points carefully. It's unlikely, though conceivable, that I missed something that should affect my ratings.

---

> ### Author Rebuttal · Authors · 2023-08-29
>
> Thanks for your approval of our work in refining problems as a whole for better reasoning performance. We also appreciate your valuable suggestions and we'll make efforts to improve the quality of the next manuscript. We hope we can address your concerns with following text.
>
> 1. **Question about the performance of Self-Polish on datasets where implicit reasoning steps are required ,e.g., MATH (Hendrycks et al.) and JEEBench ( Arora et al.).**
>
>    - Thanks for your valuable suggestion!  Our approach is equally effective for methods requiring implicit reasoning, as it can transform the implicit conditions of a problem into visible ones, making it easier for LLMs to answer. **Following your advice, we conducted preliminary experiments on the MATH dataset (Hendrycks et al.). Our approach demonstrated promising results.** Specifically, we randomly selected 200 samples for testing, and use the Chain-of-Thought as the answer-side method. Note that the few-shot prompts for CoT we used are sourced from CoTHub[1] . The results are as follows:
>
>      - | Problem Side Method | Performance |
>        | :------------------ | ----------- |
>        | No Refinement       | 21          |
>        | Zero-shot SP        | 23.5        |
>        | In-context SP       | 24.5        |
>
>    - We can observe that Self-Polish performs well on MATH. We will provide comprehensive experimental results on the implicit reasoning datasets in upcoming versions.
>
> 2. **Question about the number of LLM calls the proposed method takes on average for each dataset.**
>
>    - Thank you very much for your question. This reminds us that we haven't included all the counts of LLM calls in the paper. **In fact, in Figure 5, we showcase the average iterating times required for problem refinement on the gsm8k dataset**, with different settings of the maximum iterating times (i.e., T). For CoT-series and Standard few-shot answer-side methods, suppose it takes m iterating times, then the required LLM calls are calculated as follows:
>
>      - (the first call to answer the initial problem) + (m times problem refinement) + (m times answering new problems) = 2m + 1.
>
>    - Thus, when T is set to 2, Figure 5 indicates that the average count of LLM calls stands at 4 for Std, and 3.78 for CoT.  **Here, we list the required count of LLM calls for different datasets:**
>
>    - | Answer side | Problem side  | GSM8K | AQuA | SVAMP | MultiArith | MathQA |
>      | ----------- | ------------- | ----- | ---- | ----- | ---------- | ------ |
>      | Std         | In-context SP | 4.00  | 3.48 | 3.30  | 3.66       | 3.74   |
>      | Std         | Auto-SP       | 4.02  | 3.48 | 3.40  | 3.72       | 3.74   |
>      | Std         | Complex-SP    | 3.92  | 3.38 | 3.32  | 3.80       | 3.74   |
>      | CoT         | In-context SP | 3.78  | 3.84 | 3.32  | 3.28       | 4.02   |
>      | CoT         | Auto-SP       | 3.82  | 3.82 | 3.46  | 3.28       | 4.00   |
>      | CoT         | Complex-SP    | 3.72  | 3.82 | 3.36  | 3.18       | 3.96   |
>      | Auto-CoT    | In-context SP | 3.80  | 3.92 | 3.38  | 3.24       | 4.02   |
>      | Auto-CoT    | Auto-SP       | 3.72  | 3.84 | 3.44  | 3.28       | 4.04   |
>      | Auto-CoT    | Complex-SP    | 3.76  | 3.72 | 3.42  | 3.20       | 3.94   |
>      | Complex-CoT | In-context SP | 3.72  | 3.84 | 3.46  | 3.20       | 4.02   |
>      | Complex-CoT | Auto-SP       | 3.70  | 3.86 | 3.52  | 3.28       | 3.98   |
>      | Complex-CoT | Complex-SP    | 3.66  | 3.86 | 3.50  | 3.20       | 3.98   |
>
>    - We can observe that different answer-side methods and different problem-side methods result in varying numbers of LLM calls. This disparity arises due to differences in their success rates in achieving convergent answers. Furthermore, different datasets also require different counts of LLM calls, which might be attributed to differences in their levels of difficulty.
>
> 3. **Question of the computational cost when compared with Self-Consistency.**
>
>    - Thank you very much for your question. Firstly, self-polish and self-consistency are methods from two different sides (problem side and answer side) with the common goal of enhancing the reasoning performance of LLMs. **Compared to answer-side methods, the process of problem refinement naturally increases** **LLM** **calls, which is a cost incurred for exploring higher reasoning performance.** In fact, self-polish and self-consistency are **orthogonal**. When we combine methods from both the problem and answer sides, in the setting you mentioned, the results can reach **66.1 and 65.2**, as shown in Table 3. Hence, rather than pitting methods from different sides against each other, **combining approaches from the answer and problem sides should be pursued for achieving enhanced effectiveness.**
>    - Secondly, in the setting you mentioned, the improvement brought by Self-Polish might appear relatively limited compared to self-consistency possibly **due to the inherent high quality of the problems in standard benchmark**. However, in real-world applications, the problems LLMs encounter are often ambiguous, verbose, lacking clear focus, or include irrelevant information. These challenges could potentially mislead the model, as demonstrated in [2].
>    - In this work, our main contribution lies in introducing the idea of problem-side refinement. Of course, we could also explore the design of prompts that combine problem refinement and answering within a same LLM call to save computational costs. We leave this for future research.
>
> [1] Fu, Yao, et al. "Chain-of-Thought Hub: A Continuous Effort to Measure Large Language Models' Reasoning Performance." arXiv preprint arXiv:2305.17306 (2023).
>
> [2] Shi, Freda, et al. "Large language models can be easily distracted by irrelevant context." ICML 2023

---

### Meta-Review · Area_Chair_hUym · 2023-09-25

**Recommendation:** 4

**Metareview:**

The paper proposes a simple yet effective idea of problem-side refinement as a means to facilitate LLMs’ reasoning ability by refining the problems to be more solvable. The strategy seems effective when used in combination with Chain-of-thought prompting. While this is primarily a prompt engineering contribution, I believe it can be useful and impactful like the CoT idea. The author responses to reviewers’ concerns seemed reasonable and convincing to me.

There are concerns about the novelty of specific sub-ideas within this idea, and about missing discussion on performance differences between different variants of the proposed idea. The latter particularly can be useful to enhance the paper.

---

### Decision · Program_Chairs · 2023-10-07

**Decision:**

Accept-Findings

**Comment:**

The paper proposes a simple yet effective idea of problem-side refinement as a means to facilitate LLMs’ reasoning ability by refining the problems to be more solvable. The strategy seems effective when used in combination with Chain-of-thought prompting. While this is primarily a prompt engineering contribution, I believe it can be useful and impactful like the CoT idea. The author responses to reviewers’ concerns seemed reasonable and convincing to me.

There are concerns about the novelty of specific sub-ideas within this idea, and about missing discussion on performance differences between different variants of the proposed idea. The latter particularly can be useful to enhance the paper.